# Activation Mechanisms and Diverse Functions of Mammalian Phospholipase C

**DOI:** 10.3390/biom13060915

**Published:** 2023-05-31

**Authors:** Kaori Kanemaru, Yoshikazu Nakamura

**Affiliations:** Department of Applied Biological Science, Faculty of Science and Technology, Tokyo University of Science, Chiba 278-8510, Japan

**Keywords:** phospholipase C, phosphatidylinositol 4,5-bisphosphate, inositol 1,4,5-trisphosphate, diacylglycerol

## Abstract

Phospholipase C (PLC) plays pivotal roles in regulating various cellular functions by metabolizing phosphatidylinositol 4,5-bisphosphate in the plasma membrane. This process generates two second messengers, inositol 1,4,5-trisphosphate and diacylglycerol, which respectively regulate the intracellular Ca^2+^ levels and protein kinase C activation. In mammals, six classes of typical PLC have been identified and classified based on their structure and activation mechanisms. They all share X and Y domains, which are responsible for enzymatic activity, as well as subtype-specific domains. Furthermore, in addition to typical PLC, atypical PLC with unique structures solely harboring an X domain has been recently discovered. Collectively, seven classes and 16 isozymes of mammalian PLC are known to date. Dysregulation of PLC activity has been implicated in several pathophysiological conditions, including cancer, cardiovascular diseases, and neurological disorders. Therefore, identification of new drug targets that can selectively modulate PLC activity is important. The present review focuses on the structures, activation mechanisms, and physiological functions of mammalian PLC.

## 1. Introduction

Phospholipase C (PLC) hydrolyzes phosphatidylinositol 4,5–bisphosphate (PI(4,5)P_2_) to generate two second messengers, inositol 1,4,5 triphosphate (IP_3_) and diacylglycerol (DAG) [1,2], enabling eukaryotic cells to perform diverse functions such as cell proliferation, differentiation, and motility by spatially and temporally activating phosphoinositide turnover. Mammals possess 13 typical PLC isozymes, which can be categorized into six classes: PLCβ (β1–β4), PLCγ (γ1 and γ2), PLCδ (δ1, δ3, and δ4), PLCε, PLCζ, and PLCη (η1 and η2) [3,4,5]. The seventh family of PLC, referred to as PLCXD, has been identified in various eukaryotic species [6]. Thus, the PLC superfamily in mammalian cells comprises 16 members, with three PLCXDs (PLCXD1, PLCXD2, and PLCXD3). While it remains unclear why there is a need for such a multitude of PLC isozymes in mammalian cells despite catalyzing the same reaction, possible reasons could include distinct regulatory mechanisms and tissue distribution for each PLC isozyme (as described in Section 2 and Section 3, respectively).

Typical PLC isozymes possess a structure characterized by several conserved domains along with class-specific domains. The active sites and catalytic residues in typical PLC isozymes are located within the triosephosphateisomerase (TIM) barrel (X and Y) domains. While PLCζ is the only exception that lacks the pleckstrin homology (PH) domain, typical PLC isozymes harbor the PH domain, EF-hand motifs, and the C2 domain along with the X and Y domains. PLCβ possesses the C-terminal domain (CTD) of approximately 400 amino acids and the PSD-95, discs large, ZO-1 (PDZ)-binding motif. PLCγ bears the multidomain insertion between the X and Y domains, comprising the split PH domain, the N-terminal Src homology 2 (nSH2) domain, the C-terminal SH2 (cSH2) domain, and the Src homology 3 (SH3) domain. PLCε harbors the Cdc25 homology domain and Ras association domains. Contrary to typical PLC isozymes, the PLCXD family is a group of enzymes that contain a catalytic domain with a sequence that is similar to the X domain (Figure 1).

Upon exposure to various stimuli, typical PLC isozymes hydrolyze plasma membrane (PM) PI(4,5)P_2_ to produce two second messengers: IP_3_ and DAG [1,2]. IP_3_ binds IP_3_ receptors present in the endoplasmic reticulum (ER), inducing the release of Ca^2+^ into the cytosol from ER stores, while hydrophobic DAG binds proteins, including protein kinase C (PKC), for its membrane recruitment and activation. In addition, DAG activates transient receptor potential canonical (TRPC)3, TRPC6, and TRPC7, which are members of the TRP family of nonselective cation channels [7]. These channels are permeable to Ca^2+^ and can increase intracellular Ca^2+^ concentration. IP_3_ is metabolized to inositol 1,3,4,5-tetrakisphosphate (IP_4_) via phosphorylation by inositol 1,4,5-trisphosphate 3-kinase or inositol polyphosphate multikinase (IPMK). IP_4_ can be further metabolized to inositol 1,3,4,5,6-pentakisphosphate (IP_5_) by phosphorylation at the 6-position via IPMK and then to inositol hexakisphosphate (IP_6_) by phosphorylation at the 2-position via inositol 1,3,4,5,6-pentakisphosphate 2-kinase. IP_5_ and IP_6_ serve as substrates for the synthesis of inositol pyrophosphates (PP-InsPs) with high-energy phosphate bonds [8,9,10,11,12]. PP-InsPs are involved in various cellular processes, including chromatin remodeling, gene expression, membrane transport, insulin secretion, growth factor/cytokine signaling, apoptosis, and dopamine release [8,13,14]. In addition, IP_3_ is metabolized to inositol 1,4-bisphosphate (IP_2_) by dephosphorylation of the inositol ring at position 5 by inositol polyphosphate 5-phosphatase. IP_2_ is further dephosphorylated to myo-inositol by inositol monophosphatase or inositol polyphosphate 1-phosphatase. Myo-inositol is then re-incorporated into the phosphatidylinositol (PI) synthesis cycle by binding to CDP-DAG in the ER membrane. On the other hand, DAG is phosphorylated by DAG kinases to produce phosphatidic acid (PA) [15]. The specific acyl chain composition of PI(4,5)P_2_, with a high enrichment of stearic acid at the sn-1 position and arachidonic acid at the sn-2 position [16], is retained in the DAG generated by PLC. DAG lipases remove stearic acid, generating endocannabinoid 2-arachidonoyl glycerol, which acts as an agonist of endocannabinoid receptors [17,18]. DAG generated by the hydrolysis of PI(4,5)P_2_ is recycled into PI to maintain the total pool of phosphatidylinositol phosphates. This process involves the transport of the generated DAG and/or PA from the PM to the ER, where the PI synthetic enzymes CDP-DAG synthase and PI synthetase utilize them. This cycle is spatially confined to the PM–ER contact sites, where lipid transfer proteins transport lipid intermediates between the membranes. PLC also regulates the levels of its substrate, PI(4,5)P_2_. PI(4,5)P_2_ directly regulates various cellular functions, such as cytoskeletal remodeling, cytokinesis, phagocytosis, membrane dynamics, epithelial characterization, and ion channel activity [19,20,21,22,23]. PI(4,5)P_2_ also acts as a precursor to phosphatidylinositol 3,4,5-triphosphate (PI(3,4,5)P_3_), which triggers the activation of several other proteins, including AKT. This pathway plays a crucial role in numerous signaling processes, such as cell growth and survival [24]. Therefore, PLC-mediated hydrolysis of PI(4,5)P_2_ may exert multiple downstream effects (Figure 2).

Besides PI(4,5)P_2_, PLC enzymes have been reported to hydrolyze phosphatidylinositol 4-phosphate (PI(4)P) and, to a much lesser extent, PI in vitro [25]. Notably, PLCε could hydrolyze PI(4)P at the Golgi apparatus [26]. Several isozymes of PLC also hydrolyze nuclear PI(4,5)P_2_. Insulin-like growth factor 1 induces the activation of nuclear PLCβ1 and PI(4,5)P_2_ hydrolysis, thereby increasing nuclear DAG levels and inducing PKC nuclear translocation [27,28]. PLCβ1 isozyme has two splicing variants, PI-PLCβ1a and PI-PLCβ1b, which differ in their C-terminal sequences and intracellular localization [29]. Both variants carry a nuclear localization sequence (NLS); however, PI-PLCβ1a also possesses a nuclear export sequence (NES), allowing it to localize in the cytoplasm. Conversely, PI-PLCβ1b is primarily localized to the nucleus [30,31]. PLCγ1 induces nuclear generation of DAG [32]. PLCδ1 harbors an NES and an NLS, which contribute to nuclear–cytoplasmic shuttling [33]. Nuclear import of PLCδ1 is induced by increased cytoplasmic Ca^2+^ concentration [34]. PLCδ4 is primarily localized to the nucleus and responsible for regulating the transition between the G1 and S phases of the cell cycle [35]. PLCδ4 knockdown in adipose-derived mesenchymal stromal cells induced cell cycle arrest, with accumulation in the G1 phase [36].

## 2. Regulatory Mechanisms

Classical PLC enzymes have a shared regulatory mechanism where the enzyme’s active site is masked by the negatively charged X–Y linker and remains inactive. When PLC enzymes bind to the PM, the X–Y linker is pushed away by the negatively charged surface of the membrane, allowing the active site to become accessible and removing its auto-inhibition [37].

### 2.1. Regulatory Mechanisms of PLCβ

PLCβ isozymes act as downstream effectors of G protein-coupled receptors (GPCRs) and can be activated by either the Gαq family or Gβγ subunits [38,39]. The PH domain of PLCβ is involved in the activation of the enzyme by Gβγ and Rac [40,41]. Rac and Gβγ interact with the PH domain of PLCβ to optimize its orientation for substrate membranes [40]. PLCβ contains a CTD composed of approximately 400 amino acids, which bind to its catalytic core and inhibit enzymatic activity under resting conditions [42,43]. The CTD of PLCβ1 increases the curvature of the PM, thereby promoting efficient cleavage of PI(4,5)P_2_, which is present in highly curved membranes [44]. The activation of PLCβ by Gαq also requires the presence of a CTD. The PDZ-binding motif of PLCβ may facilitate selective binding to GPCRs via the PDZ scaffold proteins [45]. Furthermore, PLCβ functions as a GTPase-activating protein for Gαq in addition to its lipase activity [46]. Thus, PLCβ isozymes are activated by Gαq, Gβγ, and small GTPases of the Rho family, such as Rac (Figure 3).

### 2.2. Regulatory Mechanisms of PLCγ

PLCγ isoforms are regulated by both receptor tyrosine kinases (RTKs) and non-RTKs (Figure 3) [47,48,49,50,51]. Activation of PLCγ occurs via the binding of its nSH2 domain to phosphorylated tyrosine residues of RTKs, which induces the phosphorylation of a conserved tyrosine residue (Tyr783 in human PLCγ1 and Tyr759 in human PLCγ2) by RTKs [52]. The cSH2 domain inhibits PLCγ by interacting with residues around its catalytically active site under resting conditions. Phosphorylation of the conserved tyrosine residue removes the cSH2 domain from the active site via interaction with the cSH2 domain, allowing the binding of the active site of PLCγ to its substrate, PI(4,5)P_2_ [53,54]. Therefore, the PLCγ SH2 domain plays an essential role in RTK- and non-RTK-mediated PLCγ activation. PLCγ2, but not PLCγ1, interacts with Rac via the split PH domain, resulting in its recruitment to the PM and activation [55,56]. PI(3,4,5)P_3_ also recruits PLCγ isoforms to the PM and activates them [57,58,59]. Thus, the multidomain insertion located between the X and Y domains of PLCγ is essential for regulating its activity.

### 2.3. Regulatory Mechanisms of PLCδ

PLCδ activity can be stimulated by micromolar levels of Ca^2+^ within the physiological range through the activation of the other PLC isozymes or influx of Ca^2+^ through calcium channels (Figure 3) [60,61]. Ca^2+^ induces the translocation of PLCδ from the cytoplasm to the PM where it is activated. Therefore, PLCδ is thought to amplify elevated Ca^2+^ levels to concentrations sufficient for inducing downstream signaling. The PH domain also plays a critical role in activation of PLCδ. The PH domain of PLCδ binds specifically and with high affinity to PI(4,5)P_2_ [62,63], playing a crucial role in both the recruitment and activation of PLCδ on the PM. In vitro studies have suggested that the PH domain of PLCδ1 has a higher affinity for IP_3_ than for PI(4,5)P_2_ [64]. Since increased cytosolic IP_3_ levels inhibit the binding of PLCδ1 to PM PI(4,5)P_2_ [65], this may function as a negative feedback mechanism. Two putative positive regulators of PLCδ1, transglutaminase II and Ral, have been also identified [66,67].

### 2.4. Regulatory Mechanisms of PLCε

PLCε can be activated by GPCRs and RTKs, as well as by small GTPases (Figure 3) [68,69,70]. Binding to GTP-bound Rap and Ras results in differential localization of PLCε [68,71,72,73]. Ras-activating mutations and stimuli lead to PM localization of PLCε, whereas Rap activation results in its recruitment to the perinuclear region [74]. RhoA binds to PLCε through a specific region of the Y domain, resulting in its activation [75]. The Cdc25 homology domain functions as a guanine nucleotide exchange factor (GEF) for Ras and Rap1 [74,76]. The GEF activity of the Cdc25 homology domain for Rap1 can augment the lipase activity of PLCε, as activated Rap1 can stimulate PLCε. Thus, PLCε activity is regulated by various downstream signaling pathways.

### 2.5. Regulatory Mechanisms of PLCζ, PLCη, and PLCXD

PLCζ is activated by low concentrations of Ca^2+^, similar to the resting cytoplasmic Ca^2+^ concentration (Figure 3). Unlike other PLC isozymes, the X–Y linker of PLCζ exhibits distinct electrostatic features and is positively charged, which may enable it to bind to the PM or associate with the anionic substrate lipid PI(4,5)P_2._ Therefore, PLCζ is constitutively active [77]. The interaction of PLCζ with PI(4,5)P_2_ in membranes requires EF hands and the X–Y linker region, whereas its activity relies on the C2 domain [78,79].

PLCη is highly sensitive to Ca^2+^ and responds to elevated intracellular Ca^2+^ levels [80,81]. Since Gβγ also activates PLCη2, it may be activated upon GPCR stimulation (Figure 3) [82,83].

The regulatory mechanisms of PLCXDs remain unclear.

## 3. Physiological Functions of PLC

### 3.1. PLCβ

There are four isozymes of PLCβ (β1–β4), which are predominantly expressed in the brain and play essential roles in maintaining normal brain function. Several isozymes of PLCβ also play significant roles in blood cell types. PLCβ1-deficient mice experienced epileptic seizures due to impaired inhibitory neuronal circuitry; this was attributed to attenuated PKC activity, which leads to a deficit in GABAergic inhibition [84]. Similarly, human patients with PLCβ1 loss suffered from infantile epileptic encephalopathy [85,86]. In addition, PLCβ1 is crucial for glucose-stimulated insulin release in β-cells. Mice with conditional knockout (KO) of islet-expressed PLCβ1 displayed glucose intolerance, which is consistent with the observed in vitro defect [87,88]. PLCβ1 expression decreased in a malignancy-dependent manner in gliomas, and the level of PLCβ1 expression was significantly correlated with the survival rate [89]. PLCβ2 deficiency was found to inhibit Ca^2+^ release and superoxide production induced by chemoattractants in neutrophils of mice while paradoxically enhancing chemotactic activity via an unknown mechanism [90,91]. PLCβ2 also plays a central role in taste receptor signaling and is activated by βγ subunits released by various GPCRs [92,93,94]. In addition, PLCβ2 negatively regulates virus-induced pro-inflammatory responses by hydrolyzing PI(4,5)P_2_ and inhibiting PI(4,5)P_2_-mediated TGF-β-activated kinase 1 activation [95]. Loss of PLCβ3 inhibited the Src homology region 2 domain-containing phosphatase (SHP)-mediated suppression of Lyn, resulting in defective Fc epsilon Receptor I (FcεRI) signaling and mast cell-dependent immune responses in mice [96]. Loss of PLCβ3 impaired the formation of the signal transducer and activator of transcription (STAT)5–SHP-1–PLCβ3 protein complex, leading to STAT5 hyperactivation, mast cell hyperproliferation, and atopic dermatitis-like skin inflammation [97]. Interestingly, the lipase activity of PLCβ3 is not required for STAT5 regulation. In hematopoietic stem cells (HSCs), loss of PLCβ3 led to STAT5 hyperactivation, thereby increasing the number of HSCs with a myeloid differentiation ability and leading to the development of myeloproliferative neoplasms in PLCβ3-KO mice [98]. Mice lacking PLCβ3 exhibited increased sensitivity to apoptotic induction in their macrophages, which resulted in reduced atherosclerotic lesion size [99]. In humans, PLCβ3 mutations have been shown to either protect against cystic fibrosis or cause autosomal recessive spondylometaphyseal dysplasia [100,101,102]. Loss of PLCβ4 induced a range of phenotypic defects in mice, including impaired cerebellar development, which led to ataxia [103] and visual processing deficits [104]. PLCβ4 KO mice also exhibit absence seizures [105]. Studies involving human patients have shown that *PLCB4* mutations are linked to the development of uveal melanomas, which are the most common type of eye tumors arising from melanocytes of the uveal tract [106]. Loss-of-function mutations in *PLCB4* have also been implicated in auriculocondylar syndrome [107].

### 3.2. PLCγ

There are two isozymes of PLCγ (γ1 and γ2). PLCγ isozymes play essential roles in hematopoietic cell development and functions. Functional loss of PLCγ1 resulted in defective vasculogenesis and erythrogenesis, and PLCγ1-deficient mice died on embryonic day 9 [108,109]. Moreover, PLCγ1 is crucial for T-cell receptor (TCR) signaling, which is required for T-cell activation, development, and homeostasis. T-cell-specific deletion of PLCγ1 impaired the development of regulatory T cells [110]. PLCγ1 is also involved in the development of HSCs, as PLCγ1-KO cells failed to differentiate into hematopoietic cells in PLCγ1-KO chimeric mice [111]. Additionally, PLCγ1 has been implicated in various cancers in a number of studies, and these studies have highlighted the role of PLCγ1 in tumor progression and metastases [112,113,114,115,116,117,118]. Somatic mutations in *PLCG1* have been reported in angiosarcoma [119]. PLCγ1 mutant plays a role in angiosarcoma by promoting invasiveness and influencing angiogenesis through vascular endothelial growth factor (VEGF) signaling [119,120,121,122]. *PLCG1* mutations were also discovered in T-cell lymphomas, including cutaneous and adult T-cell leukemia/lymphoma. *PLCG1* is the most commonly mutated gene in adult T-cell leukaemia/lymphoma, accounting for approximately 40% of all cases. Mutant forms of this isozyme are thought to contribute to the development of cancer by promoting phospholipase activity and subsequently enhancing nuclear factor of activated T-cells (NFAT)- and NF-κB-dependent transcription [123,124]. Mutations in the TCR signaling components and *PLCG1* have been observed in T-cell lymphoma patients, and these mutations are associated with poorer overall and progression-free survival rates based on several clinical studies [125,126,127]. In contrast, some studies have indicated that reduced PLCγ1 expression is conducive to cancer cell survival and proliferation. For instance, PLCγ1 expression is downregulated during hypoxia in *KRAS*-mutant human lung adenocarcinoma cell lines, preventing lipid peroxidation, inhibiting apoptosis, and enhancing cancer cell proliferation [128]. Mice with a specific PLCγ1 KO in neuronal precursors exhibited deficiencies in midbrain axon guidance, resulting in structural alterations to the mesencephalic dopaminergic system, wherein axons fail to project to their appropriate locations [129,130,131]. Forebrain-selective PLCγ1 KO resulted in behavioral abnormalities such as hyperactivity [132]. Activation of PLCγ1 is triggered by the activation of tropomyosin-related kinase B (TrkB) receptors through binding to brain-derived neurotrophic factor (BDNF), which plays an essential role in the formation and function of inhibitory synapses that use gamma-aminobutyric acid (GABA) as a neurotransmitter. Selective PLCγ1 KO in inhibitory GABAergic neurons increased seizure susceptibility in aged mice [133]. In contrast, in a temporal lobe epilepsy model, hyperexcitation of excitatory neurons triggered the activation of cellular signaling pathways, including elevated phosphorylation of PLCγ1 via the BDNF-TrkB pathway. Uncoupling of the BDNF receptor TrkB from PLCγ1 prevented epilepsy, suggesting that the effects of PLCγ1 on epilepsy depend on the specific neuronal population involved [134]. PLCγ2 is a critical signaling effector of the pre-B-cell receptor and essential for B-cell development and maturation. PLCγ2-KO mice showed impaired B-cell maturation [135], whereas a gain-of-function mutation of PLCγ2 generated via ENU mutagenesis resulted in the hyperactivation of B-cells and innate immune cells [136]. Furthermore, PLCγ2 plays a role in the regulation of innate immune cells and platelets through the signaling of Fc receptors [137]. PLCγ2 deficiency also impaired receptor activator of NF-κB ligand (RANKL) signaling in hematopoietic cells, leading to defects in lymph node organogenesis and osteoclast differentiation [138]. Gain-of-function mutations in *PLCG2* have been linked to a disorder called PLCγ2-associated antibody deficiency and immune dysregulation (PLAID), which is characterized by cold urticaria due to the spontaneous activation of mast cells expressing the mutant form of PLCγ2 when exposed to lower temperatures [139]. In addition, gain-of-function mutations in *PLCG2* have been implicated in a complex immune disorder called autoinflammation, antibody deficiency, and immune dysregulation, which are predominantly inherited and resemble PLAID [140,141,142].

### 3.3. PLCδ

There are three isozymes of PLCδ (δ1, δ3, and δ4), which play critical roles in the normal function of the skin, osomosensitive neurons, placenta, heart, and sperm. Mice lacking PLCδ1 displayed sparse hair owing to an abnormal hair shaft structure and reduced hair keratin expression [143,144]. In addition, PLCδ1 plays a critical role in nail formation, as demonstrated by mutations in patients with hereditary leukonychia [145,146,147,148]. Furthermore, PLCδ1 is involved in the regulation of inflammatory skin diseases, such as psoriasis and contact hypersensitivity (CHS) [149,150]. PLCδ1 also plays a key role in the activation of deltaN-TRPV1 channels and osmosensory transduction in magnocellular neurosecretory cells [151,152]. Epigenetic silencing of *PLCD1* has been observed in several cancers, suggesting its potential tumor-suppressive role [153,154,155,156]. Simultaneous loss of PLCδ1 and PLCδ3 in mice led to embryonic lethality due to decreased placental vascularization and excessive apoptosis of placental trophoblasts [157]. PLCδ1/PLCδ3 double-KO mice also exhibited impaired cardiac function, fibrosis, and spontaneous cardiac hypertrophy, possibly caused by excessive apoptosis of cardiomyocytes [158]. Male infertility is observed in mice lacking PLCδ4 due of their inability to initiate the acrosome reaction, which is essential for sperm penetration into the zona pellucida and fusion with the egg PM [159,160].

### 3.4. PLCε

There is only one isozyme of PLCε. Consistent with its high expression in cardiac tissues, PLCε plays a critical role in the regulation of cardiomyocyte development and function. Increased PLCε transcript levels were observed in the myocardial tissues of patients with idiopathic dilated cardiomyopathy, suggesting the potential involvement of PLCε in the pathogenesis of human cardiac diseases [161]. Studies on cardiomyocyte-specific PLCε-KO mice have demonstrated protection against pressure overload-induced hypertrophy. Mechanistically, PLCε catalyzes the hydrolysis of the noncanonical substrate PI(4)P in the perinuclear Golgi apparatus to generate DAG in cardiomyocytes. Subsequently, DAG activates the hypertrophic kinase protein kinase D [26,162]. PLCε also participates in cardiac development, as shown in mice lacking catalytically active PLCε displaying impaired cardiac semilunar valvulogenesis [163]. Furthermore, PLCε has been implicated in skin inflammation. PLCε overexpression in keratinocytes induced psoriasis-like skin inflammation [164], whereas lack of PLCε attenuated CHS [165]. Hence, PLCε appears to positively regulate skin inflammation. PLCε also plays a positive role in neuroinflammation [166]. Mutations in the X domain of *PLCE1* in humans can lead to nephrotic syndrome, characterized by proteinuria due to disruption of the glomerular filtration barrier executed by podocytes [167,168].

### 3.5. PLCζ

PLCζ, which is specifically expressed in the sperm, plays a pivotal role in fertilization. It is a key molecule derived from sperm that induces Ca^2+^ oscillation, which is a crucial process for egg activation during fertilization [169]. Studies demonstrated that PLCζ downregulation in mouse sperm impaired Ca^2+^ oscillations and egg activation [170,171]. Conversely, broad ectopic PLCζ expression led to autonomous Ca^2+^ oscillations in unfertilized oocytes, resulting in egg activation and parthenogenetic development, highlighting the direct effect of PLCζ, which is analogous to fertilization [172]. In humans, loss-of-function mutations in many PLCζ variants found in patients have been identified and linked to the failure of oocyte activation, which is regulated by Ca^2+^ oscillations [173,174,175].

## 4. Chemical Inhibitors and Activators for PLC

There are several compounds that are known to modulate the activity of PLC. U73122 is a commonly used inhibitor of PLC, although it has been reported to affect other targets, such as ion channels, calcium pumps, and enzymes [176,177,178]. Similarly, m-3M3FBS is a commonly used pan-PLC activator; however, it interacts with unrelated targets in cells and there is no clear evidence that it directly binds PLC [179]. Thus, currently, no fully validated small-molecule inhibitors or activators of PLC suitable for research applications are available. This limitation is largely due to the lack of a powerful high-throughput screening system and difficulties associated with generating chemical probes based on the PLC substrate, PI(4,5)P_2_. Recent advances have been achieved to overcome these challenges. Although the half-life of IP_3_, a direct product of PLC, is short, the downstream metabolite IP_1_ can be stabilized by introducing lithium chloride (LiCl). Therefore, PLC activity can be evaluated by measuring IP_1_ accumulation in the presence of LiCl [180]. Furthermore, there is potential for in vitro assays utilizing PLC, as demonstrated by the use of fluorescently tagged PI(4,5)P_2_ analogs such as WH-15, which can be hydrolyzed by PLC isozymes to produce a fluorescent molecule [181]. A related compound, XY-69, has also been synthesized and used in vitro [182]. Recent advances in the development of high-throughput screening systems are expected to facilitate the identification of specific PLC inhibitors and activators.

## 5. Perspectives

PLC exerts its physiological functions primarily through generation of the second messengers IP_3_ and DAG. However, considering the involvement of PI(4,5)P_2_ in the regulation of diverse cellular functions, the reduction in PI(4,5)P_2_ levels caused by PLC is highly likely to play a role in its physiological functions. Further investigations are warranted to determine the impact of PLC-mediated PI(4,5)P_2_ metabolism on PI(4,5)P_2_ levels in various cellular contexts. Besides enzymatic activity, some PLC isozymes have multifunctional roles. For instance, PLCβ1 regulates caveolar invasion and membrane curvature in a lipase-independent manner. Future studies should explore the lipase-independent functions of PLC to elucidate their novel roles. Moreover, since the structure of PLCXD is distinct from that of typical PLC, investigation concerning its substrate specificity and activation mechanism would be intriguing. In addition, specific PLC isozymes that play critical roles in certain organs may serve as viable targets for the development of novel drugs. Structural data on PLC isozymes and the availability of fluorescent substrates can allow for the screening of specific PLC activators and inhibitors as potential drug candidates.

## Figures and Tables

**Figure 1 biomolecules-13-00915-f001:**
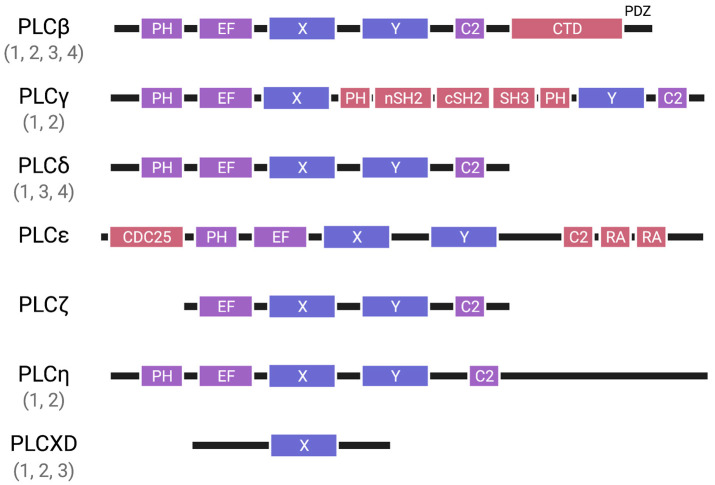
The domain structures of PLC. The active sites and catalytic residues in typical PLC isozymes are located within the X and Y domains (X and Y). While PLCζ is the only exception that lacks the PH domain, all typical PLC isozymes also possess the PH domain (PH), EF-hand motifs (EF), and the C2 domain (C2). PLCβ has the C-terminal domain (CTD), as well as the PDZ-binding motif. PLCγ features a multidomain insertion between the X and Y domains, consisting of the split PH domain (PH), N-terminal SH2 domain (nSH2), C-terminal SH2 domain (cSH2), and SH3 domain (SH3). PLCε has Ras association domains (RA) and a Cdc25 homology domain (CDC25). Atypical PLC isozymes, the PLCXD family contain a catalytic domain with a sequence that is similar to the X domain (X).

**Figure 2 biomolecules-13-00915-f002:**
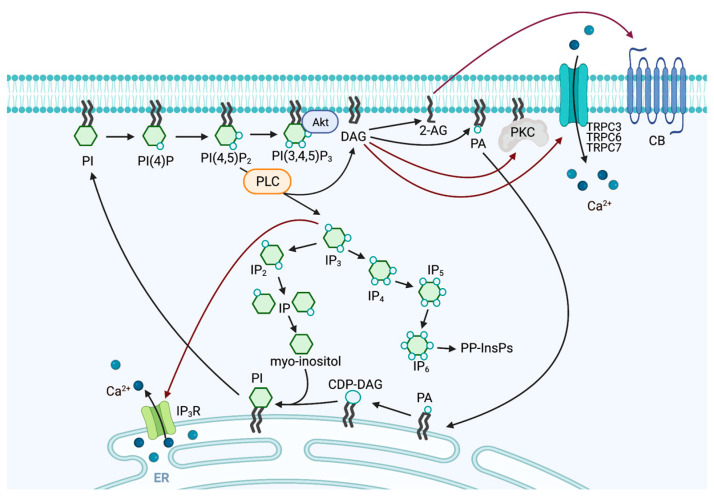
The schematic pathway of PI turnover induced by PLC. PLC isozymes hydrolyze PM PI(4,5)P_2_ to produce two second messengers: IP_3_ and DAG. IP_3_ binds IP_3_ receptors (IP_3_R) present in the ER, inducing the release of Ca^2+^ into the cytosol, while DAG activates PKC. DAG also activates TRPC3, TRPC6, and TRPC7 and increases intracellular Ca^2+^ concentration. IP_3_ is metabolized to IP_4_, IP_5_, IP_6_, and PP-InsPs. IP_3_ is also metabolized to IP_2_. IP_2_ is further dephosphorylated to myo-inositol. Myo-inositol is then re-incorporated into the PI synthesis cycle by binding to CDP-DAG in the ER membrane. DAG is metabolized to PA. DAG and PA are transported from the PM to the ER, where the PI synthetic enzymes utilize them. Thus, DAG generated by the hydrolysis of PI(4,5)P_2_ is recycled into PI to maintain the total pool of phosphatidylinositol phosphates. DAG is metabolized to endocannabinoid 2-arachidonoyl glycerol (2-AG), which acts as an agonist of endocannabinoid receptors (CB). PI(4,5)P_2_ is also metabolized to PI(3,4,5)P_3_, which activates AKT.

**Figure 3 biomolecules-13-00915-f003:**
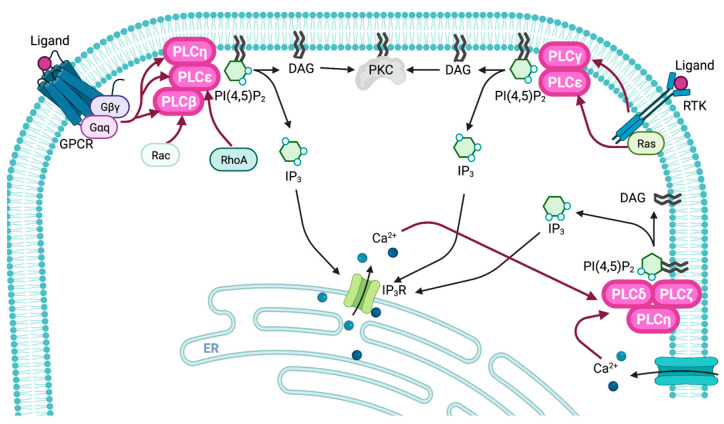
Activation mechanisms of PLC. PLCβ isozymes are activated by Gαq, Gβγ, and Rac. PLCγ isoforms are activated by RTKs. PLCδ activity can be stimulated by Ca^2+^ within the physiological range through the activation of the other PLC isozymes or influx of Ca^2+^ through calcium channels. PLCε can be activated by GPCRs and RTKs as well as by small GTPases. PLCζ and PLCη are highly sensitive to Ca^2+^ and respond to small elevations in intracellular Ca^2+^ levels. PLCη is also activated by GPCR.

## Data Availability

Not applicable.

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
