# Peer review of "Activation Mechanisms and Diverse Functions of Mammalian Phospholipase C"

_biomolecules, 2023, doi:10.3390/biom13060915_

Round 1
Reviewer 1 Report
In this manuscript, Kanemaru and Nakamura thoroughly describe the mammalian PLC family, including a detailed report on their activation mechanisms and functions. The literature review is quite extensive, and it gives a good idea of the available information. The figures are appropriate and nicely prepared. The subject is timely and properly deserves publication in Biomolecules.
I do have, however, some comments that first need to be addressed.
In my opinion, the weak point of the manuscript is the introduction. Although the entire manuscript would benefit from some editing (kindly see examples below), the introduction portrays the information in a way that is not so easy to follow. It is true that the actual content is quite thorough, but the transitions between sentences and the existence of only 2 paragraphs turns the entire section complicated to read and properly understand. Re-writing this section would greatly improve the manuscript, as well as make it more appealing to a broader audience.
Minor comments (and examples of required editing/re-writing):
- (Introduction) To help readers less familiar with PLC-related biology, a couple of sentences should be added to explore a bit why are there so many classes within mammals (for example, PLCz is specific to sperm, and so on). I also think it would be much simpler to have the hydrolytic function stated in a sentence right in the beginning of the introduction, to contextualize.
- (Page 1, line 33) “Except PLCz, all typical PLC 33 isozymes harbor the pleckstrin homology (PH) domain, EF-hand motifs, and the C2 34 domain, along with the X and Y domains.” Although it is clear in the figure, this sentence seems to suggest that PLCz does not have any of the stated domains. Please re-phrase.
- The use of words as “additionally” and “moreover” is quite extensive and should be reduced in the final version.
- (Page 4, line 127) “The PH domain of 127 PLCb is involved in the activation of the enzyme by Gbg, Rac, and Cdc42 [40,41].” Can you please expand on how it is involved?
- Figure 3 would benefit from PLC variants being more visible. In other words, although the schematic representation is clear, and since the review is on PLC, it would be nice that the PLCs stand out and readers can immediately find what they are looking for.
- In section 3, subsections should start with a brief and very general description of each class’ overall functions, again to make it clearer. For example, section 3.1 starts with a practical and very specific example, making it more difficult to follow.
- (Section 2.3) Just as an additional example of the necessary editing, the following sentences appear within five consecutive lines: “Ca2+ induces the translocation of PLCd from the cytoplasm to the PM, 163 where it is activated.” (…) “This interaction recruits PLCd to the 166 PM, where it is activated.”
Overall, this is a very thorough revision of PLC’s activation and function, and it is worth publishing in Biomolecules upon some alterations. The implementation of the above-mentioned comments will improve the readability and make it more accessible to students, non-PLC experts and, in general, to a broader audience.
The required text editing is mentioned in the above comments.
Author Response
Major comment
In my opinion, the weak point of the manuscript is the introduction. Although the entire manuscript would benefit from some editing (kindly see examples below), the introduction portrays the information in a way that is not so easy to follow. It is true that the actual content is quite thorough, but the transitions between sentences and the existence of only 2 paragraphs turns the entire section complicated to read and properly understand. Re-writing this section would greatly improve the manuscript, as well as make it more appealing to a broader audience.
We appreciate comments regarding the introduction, and we have carefully considered the reviewer’s suggestions. We have made revisions to the introduction section to address the issues the reviewer raised. Firstly, we have divided the introduction into four paragraphs to enhance the readability and improve the flow of information. This division allows for a more organized presentation of the content, making it easier for readers to follow and understand. Secondly, we have added a description of the enzymatic reaction catalyzed by PLC at the beginning of the introduction (line 25-29). This addition provides a contextual background for readers less familiar with PLC-related biology, allowing them to grasp the fundamental process being discussed. Finally, we have included an explanation as to why there are so many PLC isozymes. We approached this explanation based on tissue distribution and activation mechanisms (line 34-37). We believe that these revisions have significantly improved the introduction section by addressing the issues the reviewer raised and enhancing its overall clarity and accessibility.
Minor comments (and examples of required editing/re-writing):
(Introduction) To help readers less familiar with PLC-related biology, a couple of sentences should be added to explore a bit why are there so many classes within mammals (for example, PLCz is specific to sperm, and so on). I also think it would be much simpler to have the hydrolytic function stated in a sentence right in the beginning of the introduction, to contextualize.
Please see our response to the major comment.
(Page 1, line 33) “Except PLCz, all typical PLC 33 isozymes harbor the pleckstrin homology (PH) domain, EF-hand motifs, and the C2 domain, along with the X and Y domains.” Although it is clear in the figure, this sentence seems to suggest that PLCz does not have any of the stated domains. Please re-phrase.
Thank you for pointing out the potential confusion in the sentence regarding PLCζ and the mentioned domains. We appreciate the reviewer’s suggestion for rephrasing the sentence to avoid any misunderstandings. We have revised the sentence as follows: "While PLCζ is the only exception that lacks the pleckstrin homology (PH) domain, typical PLC isozymes harbor the PH domain, EF-hand motifs, and the C2 domain, along with the X and Y domains."(line 41-43).
The use of words as “additionally” and “moreover” is quite extensive and should be reduced in the final version.
Thank you for bringing up the issue regarding the extensive use of the words "additionally" and "moreover" in the manuscript. We have carefully reviewed the text and made significant revisions to reduce the frequency of these terms.
(Page 4, line 127) “The PH domain of PLCb is involved in the activation of the enzyme by Gbg, Rac, and Cdc42 [40,41].” Can you please expand on how it is involved?
Thank you for your comment regarding the involvement of the PH domain of PLCβ in the activation of the enzyme by Gβγ, Rac, and Cdc42. In response to the reviewer’s feedback, we have added the following sentence to the manuscript: "Rac and Gβγ interact with the PH domain of PLCβ to optimize its orientation for substrate membranes."(line 167-169). Furthermore, considering the varying reports on the involvement of Cdc42 in the regulation of PLCβ activity, we have decided to remove references to Cdc42 from both the manuscript and Figure 3. This adjustment ensures that the content accurately reflects the current understanding of PLCβ activation mechanisms.
Figure 3 would benefit from PLC variants being more visible. In other words, although the schematic representation is clear, and since the review is on PLC, it would be nice that the PLCs stand out and readers can immediately find what they are looking for.
Thank you for the valuable feedback regarding Figure 3. We have carefully considered the reviewer’s suggestion and made the necessary modifications to enhance the visibility of PLC variants in the figure. In response to the reviewer’s comment, we have made the PLC variants more prominent and easily identifiable within the schematic representation. We have adjusted the design elements such as color, size, and labeling to ensure that the PLCs stand out and are readily noticeable to readers.
In section 3, subsections should start with a brief and very general description of each class’ overall functions, again to make it clearer. For example, section 3.1 starts with a practical and very specific example, making it more difficult to follow.
Thank you for the valuable feedback regarding the organization of subsections in Section 3. In response to the reviewer’s comment, we have added a brief introductory statement at the beginning of each subsection in Section 3, highlighting contributions of each PLC class in maintaining the functions of various tissues (line 248-250, 285-286, 345-346, and 365). These additions aim to provide readers with a broader understanding of the functional relevance of each PLC class before delving into more specific examples and details.
(Section 2.3) Just as an additional example of the necessary editing, the following sentences appear within five consecutive lines: “Ca2+ induces the translocation of PLCd from the cytoplasm to the PM, 163 where it is activated.” (…) “This interaction recruits PLCd to the 166 PM, where it is activated.”
We appreciate the reviewer’s feedback, and we have made the necessary changes to clarify the important roles of Ca2+ and the PH domain in the activation of PLCδ. In response to the reviewer’s comment, we have revised the relevant sentences to explicitly state the roles of Ca2+ and the PH domain in the activation process of PLCδ. (line 210-213).
Overall, this is a very thorough revision of PLC’s activation and function, and it is worth publishing in Biomolecules upon some alterations. The implementation of the above-mentioned comments will improve the readability and make it more accessible to students, non-PLC experts and, in general, to a broader audience.
We have carefully considered all reviewer's comments and suggestions, and we have implemented the necessary changes to address the reviewer’s concerns. We would like to express our gratitude to the reviewer for their valuable input, which has undoubtedly enhanced the quality and clarity of the manuscript.
Reviewer 2 Report
This is comprehensive review article by Kanemaru and Nakamura. This review nicely summarized the molecular mechanisms, biological and physiological function and small molecule inhibitors and activators of PLC family members. The manuscript was well written, and the sections were logically arranged. The complex mechanisms have been clearly and concisely demonstrated. Figures were also effective in illustrating the mechanisms of PLC family members. In summary, this is a very nice review. Except some minor abbreviation issues, I recommend publication of this review.
Abbreviation: TIM (line 33), EF (line 34), PDZ (line 36), SH2 (line 38), SH3 (line 48), SHP (line 209), STAT5 (line 211), FcεRI (line 210), knockout (add “KO” to line 199), PLCG1 (line 237), VEGF (line 238), NFAT (line 244), TrkB (line 256), RANKL (line 270), PLCE (line 310).
Author Response
This is comprehensive review article by Kanemaru and Nakamura. This review nicely summarized the molecular mechanisms, biological and physiological function and small molecule inhibitors and activators of PLC family members. The manuscript was well written, and the sections were logically arranged. The complex mechanisms have been clearly and concisely demonstrated. Figures were also effective in illustrating the mechanisms of PLC family members. In summary, this is a very nice review. Except some minor abbreviation issues, I recommend publication of this review.
Abbreviation: TIM (line 33), EF (line 34), PDZ (line 36), SH2 (line 38), SH3 (line 48), SHP (line 209), STAT5 (line 211), FcεRI (line 210), knockout (add “KO” to line 199), PLCG1 (line 237), VEGF (line 238), NFAT (line 244), TrkB (line 256), RANKL (line 270), PLCE (line 310).
Thank you very much for comprehensive review and positive feedback on our manuscript. We are glad that the reviewer found our review well-written and the sections logically arranged. We appreciate the reviewer’s recognition of our efforts in summarizing the molecular mechanisms, biological and physiological functions, as well as small molecule inhibitors and activators of PLC family members.
We acknowledge the abbreviation issues mentioned in the reviewer’s comments, and we will make the necessary revisions to address them accordingly. Specifically, we will expand "TIM" to "triosephosphateisomerase", "PDZ" to " PSD-95, discs-large, ZO-1", "SH2" to "Src homology 2," "SH3" to "Src homology 3," "SHP" to "Src homology region 2 domain-containing phosphatase," "STAT5" to "signal transducer and activator of transcription 5," "FceRI" to " Fc epsilon Receptor I", "VEGF" to "vascular endothelial growth factor," "NFAT" to "nuclear factor of activated T cells", "TrkB" to "tropomyosin receptor kinase B", "RANKL" to "receptor activator of NF-kB ligand". Since we cannot find the way to spell out for EF, and PLCG1 and PLCE are gene names, we did not spell them out.
We appreciate your recommendation for the publication of our review, and we will carefully consider the reviewer’s valuable feedback to further enhance the quality and clarity of the manuscript. Thank you once again for the reviewer’s thoughtful evaluation.
Reviewer 3 Report
The proposed work is a review about mammalian phospholipases C (PLC). The paper presents a classification of PLCs based on their structure. The paper describes in detail the mechanisms that regulate the work of the PLC and the features of the functioning of phospholipases of various classes. The manuscript is well written and contains a lot of information that can be useful in the development of PLC-specific drugs that regulate its work.
I recommend to accept the manuscript as is.
Author Response
The proposed work is a review about mammalian phospholipases C (PLC). The paper presents a classification of PLCs based on their structure. The paper describes in detail the mechanisms that regulate the work of the PLC and the features of the functioning of phospholipases of various classes. The manuscript is well written and contains a lot of information that can be useful in the development of PLC-specific drugs that regulate its work.
I recommend to accept the manuscript as is.
Thank you very much for review and positive feedback on our manuscript. We are glad that the reviewer found our paper well-written and informative. We would like to express our gratitude for the reviewer’s time and valuable feedback.